# CORTICALLY MOTIVATED RECURRENCE ENABLES TASK EXTRAPOLATION

## ABSTRACT

Feedforward deep neural networks have become the standard class of models in the field of computer vision. Yet, they possess a striking difference relative to their biological counterparts which predominantly perform "recurrent" computations. Why do biological neurons evolve to employ recurrence pervasively? In this paper, we show that a recurrent network is able to flexibly adapt its computational budget during inference and generalize within-task across difficulties. Simultaneously in this study, we contribute a recurrent module we call LocRNN that is designed based on a prior computational model of local recurrent intracortical connections in primates to support such dynamic task extrapolation. LocRNN learns highly accurate solutions to the challenging visual reasoning problems of Mazes and PathFinder that we use here. More importantly, it is able to flexibly use less or more recurrent iterations during inference to zero-shot generalize to less- and more difficult instantiations of each task without requiring extra training data, a potential functional advantage of recurrence that biological visual systems capitalize on. Feedforward networks on the other hand with their fixed computational graphs only partially exhibit this trend, potentially owing to image-level similarities across difficulties. We also posit an intriguing tradeoff between recurrent networks' representational capacity and their stability in the recurrent state space. Our work encourages further study of the role of recurrence in deep learning models – especially from the context of out-of-distribution generalization & task extrapolation – and their properties of task performance and stability.

## 1 INTRODUCTION

Deep learning based models for computer vision have recently matched and even surpassed human-level performance on various semantic tasks (Dosovitskiy et al., 2020; Vaswani et al., 2021; He et al., 2021; Liu et al., 2022). While the gap between human and machine task performance has been diminishing with more successful deep learning architectures, differences in their architectures and in critical behaviors such as adversarial vulnerability (Athalye et al., 2018), texture bias (Geirhos et al., 2018), lack of robustness to perceptual distortions (Hendrycks & Dietterich, 2019; Geirhos et al., 2019), etc. have increased significantly. We are interested in one such stark architectural difference between artificial and biological vision that we believe underlies the above-mentioned critical behavioral differences, i.e., recurrent neural processing. While biological neurons predominantly process input stimuli with recurrence, existing high-performing deep learning architectures are largely feedforward in nature. In this work, we argue for further incorporation of recurrent processing in future deep learning architectures – we compare matched recurrent and feedforward networks to show how the former are *capable of extrapolating representations* learned on a task to unseen difficulty levels without extra training data; feedforward networks are strictly restricted in this front as they cannot dynamically change their computational graph.

Ullman (1984) introduced a particularly important research direction in visual cognition that is of fundamental importance to understanding the human ability to extrapolate learned representations within-task across difficulties. Ullman hypothesized that all visual tasks we perform are supported by combinations of a small set of key elemental operations that are applied in a sequential manner (analogous to recurrent processing) like an instruction set. Instances of varying difficulty levels of a task can be solved by dynamically piecing together shorter or longer sequences of operations corresponding to that task. This approach of decomposing tasks into a sequence of elemental opera-

tions (aka visual routines) avoids the intractable need for vast amounts of training data representing every instantiation of a task and is hypothesized to support our human visual system's systematic generalization ability. Examples of such elemental operations that compose visual routines include incremental contour grouping, curve tracing, etc. and a review of these operations along with physiological evidence can be found in Roelfsema et al. (2000). Can we develop artificial neural networks that also learn visual routines for any task when constrained to deliberately use specialized recurrent operations?

In this work we show promising evidence for employing recurrent architectures to learn such general solutions to visual tasks and exhibit task extrapolation to various difficulty levels. We note that such extrapolation is one kind of out-of-distribution generalization that standard feedforward deep learning models struggle to perform. For this study, we perform experiments using two challenging synthetic visual tasks, namely Mazes and PathFinder (discussed in Sec. 3). We make the following contributions as part of this work:

1. We show the advantage of recurrent processing over feedforward processing on task extrapolation. We **show evidence for strong visual task extrapolation using specialized recurrent-convolutional architectures including our proposed recurrent architecture, LocRNN** on challenging visual reasoning problems.

2. We contribute LocRNN, a biologically inspired recurrent convolutional architecture that is developed based on Li (1998), a prior computational neuroscience model of recurrent processing in primate area V1. LocRNN introduces one connection type that is missing from commonly used deep neural networks – long-range lateral connections that connect neurons within the same layer in cortex Bosking et al. (1997). We hypothesize such lateral recurrent connections to enable the learning and sequencing of elemental visual routines.

3. Comparing task performance alongside extrapolation performance of our various recurrent architectures, we posit the potential *tradeoff between task performance* – the ability of recurrent networks to learn sophisticated iterative functions of their input to excel on downstream tasks – *vs stability* in the state space – local smoothness of the trajectory of recurrent states through time. We show empirical evidence for an instance of this tradeoff we observe and identify an important open problem which must be solved to establish high-performing and stable recurrent architectures for vision.

We combine cognitive insights from Ullman routines (Ullman, 1984) with a model of cortical recurrence (Li et al., 2006) from computational neuroscience to improve state-of-the-art recurrent neural network based machine learning architectures. This unique synergy results in demonstrating the superior ability of recurrent networks for task extrapolation by flexibly adapting their test-time computational budget, a feat not possible for feedforward architectures. Our work encourages future work to further explore the role of recurrence in the design of deep learning architectures that behave like humans and generalize out-of-distribution.

## 2 RELATED WORK

As mentioned our work is highly relevant to the visual routines literature introduced by Ullman (1984) and further reviewed elaborately in Roelfsema et al. (2000). The core idea of visual routines that make it relevant to our studied question of task extrapolation is the flexible sequencing of elemental operations resulting in a dynamic computational graph. This idea has been studied by prior research on using recurrent neural networks to solve sequential problems. On a related note there have been several attempts to learn how much recurrent computation to use for a given input sample on a given task (Graves, 2016; Saxton et al., 2019).

The most relevant to our work is (Schwarzschild et al., 2021) where the authors evaluate the application of recurrent neural networks to generalize from easier to harder problems. Our work extends and differs from their intriguing study of recurrence in task extrapolation in the following ways: 1) While their work explores sequential task extrapolation in general with abstract problems such as Prefix Sum and solving Chess Puzzles, our work extends it to particularly focus on extrapolation in visual task learning. Hence their maze segmentation task is of relevance to us and we use the same for evaluation (while also re-implementing models used in their study). In addition, we present evaluation on the Pathfinder challenge (Linsley et al., 2018), a relatively significantly more challenging and large-scale visual reasoning task, the design of which dates back to Jolicoeur et al. (1986).

2) Schwarzschild et al. (2021) implement only a simple and weak form of recurrence realized by weight-tying, i.e., they only evaluate ResNets with weight sharing across residual blocks. In addition to such recurrent ResNets, we present analyses with highly sophisticated recurrent architectures specialized for recurrent image processing. 3) We introduce LocRNN, a high performing recurrent architecture based on prior computational models of cortical recurrence that is inductively biased to learn sequential visual routines while trained on suitable downstream tasks.

On the design of recurrent architectures, our work is loosely relevant to Eigen et al. (2013), Pinheiro & Collobert (2014) and Liao & Poggio (2016) which discuss the role of weight sharing in feedforward networks to produce recurrent processing. We are interested, however, in designing new specialized recurrent architectures that play a role both in human and machine vision. Linsley et al. (2018) developed one such recurrent architecture called hConvGRU that we also evaluate for comparison. Relative to hConvGRU (Linsley et al., 2018), LocRNN is a relatively computationaly simpler yet more high-performing model (as highlighted by matching comparisons in Sec. 1).

## 3 DATASETS

For evaluating the ability of various models in exhibiting task extrapolation, we curate two challenging visual tasks, Mazes and PathFinder, with instances at 3 parametric difficulty levels. **Both tasks involve the visual routines of marking and curve tracing** (Ullman, 1984). These datasets are inspired by prior visual psychophysics research where such tasks were used abundantly to estimate the cognitive and neural underpinnings of sequential visual processing, such as incremental grouping, structure and preference of lateral connections, etc. (Jolicoeur & Ingleton, 1991; Ullman, 1984; Li et al., 2006; Roelfsema, 2006). In the following subsections we describe the specifics of our tasks.

### 3.1 MAZES CHALLENGE - ROUTE SEGMENTATION

**Task description** Human beings are adept at solving mazes, a task that requires application of a similar serial grouping operation like PathFinder in order to discover connectivity from a starting point to the final destination of the maze amidst blocking walls. For evaluating model performance on solving mazes of varying difficulty, we use the publicly available version of the Mazes challenge developed by Schwarzschild et al. (2021). They implemented this Mazes challenge as a binary segmentation problem where models take $N \times N$ images of square-shaped mazes as input with three channels (RGB) with the start position, end position, permissible regions and impermissible regions marked in the image. The output produced by models is a binary segmentation of the route discovered by the model from the start to the end position.

**Difficulty levels:** The Mazes challenge has been designed at 3 difficulty levels, each difficulty level is parameterized by the size of the square grid that the maze is designed into. The grid size corresponding to easy(small)/medium/large(hard) mazes is 9×9/ 11×11/ 13×13 (see example images in Fig. 5 in Appendix). This dataset consists of 50,000 training images and 10,000 testing images with the following spatial resolution: Small mazes – $24 \times 24$ pixels, Medium mazes – $28 \times 28$ pixels and Large mazes – $32 \times 32$.

**Evaluation criteria:** Although Mazes is a segmentation challenge and hence, one could potentially consider partially correct routes during evaluation (for example with average of per-pixel accuracy). However this is less strict than giving each image a single binary score reflecting if *all* pixels are labeled correctly. Evaluation criteria for mazes is hence the total *% of test-set mazes completely accurately solved* at a given difficulty level.

### 3.2 PATHFINDER CHALLENGE – CURVE TRACING

**Task description** While Mazes is an interesting challenge examined in Schwarzschild et al. (2021) and Schwarzschild et al. (2022), we consider it much smaller scale (e.g. MNIST / CIFAR-10-scale) than the PathFinder challenge. In the PathFinder task introduced by Linsley et al. (2018), models are trained to identify whether two circular disks in an input stimulus form the two ends of a locally connected path made up of small "segments". Each image consists of two main connected paths $P_0$ and $P_1$ made up of locally co-circular segments . Each image also contains two circular disks each of which is placed at one of the 4 end points of $P_0$ and $P_1$. Images that contain a disk on both

ends of the same path are classified as positive, and those containing a disk on endpoints of different paths are classified as negative.

**Difficulty levels:** Pathfinder is designed at 3 difficulty levels parameterized by the length (number of segments) of the paths $P_0$ and $P_1$ mentioned above. The easiest version of pathfinder uses paths that are 9 segments long (PathFinder-9), while the medium and hard versions contain paths that are 14 (PathFinder-14) and 18 (PathFinder-18) segments long respectively (see example images in Fig. 5 in Appendix). This dataset consists of a total of 800,000 RGB images at each difficulty level, each with a spatial resolution of $150 \times 150$ pixels. There is an equal number of positive and negative instances at each difficulty level. We use 700,000 images for training and 100,000 images as the test set at each difficulty level. For evaluating zero-shot task extrapolation on PathFinder in Sec. 5.2 we used a random sample of 20,000 images from PathFinder-9 and PathFinder-18 test sets respectively.

**Evaluation criteria** Since this is a classification challenge, we use accuracy, i.e. *% correct on test-images* as the evaluation metric to rank model performance on PathFinder. Model architectures will receive a three-channel input image and process it via a stack of standard convolution/recurrent-convolution layers followed by a classification readout which outputs two scores corresponding to the two classes. Since this is a binary classification challenge with balanced classes, chance performance is 50% when models are making random predictions.

## 4 Model architectures and training

### 4.1 Formulation of LocRNN

We note that prior work has explored the development of recurrent architectures tailored for vision. Here, we introduce a similar but highly expressive (as demonstrated by Fig. 1) recurrent architecture designed ground-up based on a computational model of iterative contour processing in primate vision from Li (1998). This model is an ODE-based computational model of interactions between cortical columns in primate area V1 mediated by "lateral connections" local to a cortical area. Analogous to the cortical area V1 is a set of activation maps in our deep networks, say, output by the first convolution layer, comprised of neurons firing in response to a variety of low-level stimulus features such as oriented bars and gradients. Distant neurons in these activation maps do not interact as interactions are restricted to the small receptive field sizes of convolutions. LocRNN symbolizes an RNN enabling long-range local interactions – iterative interactions between distant neurons that are local to a set of activation maps in a given layer of a deep network.

By discretizing the continuous-form ODE dynamics of processing units from Li (1998), we arrived at an interpretable and powerful set of dynamics for "LocRNN". The effective receptive field of LocRNN output neurons increases linearly with recurrent timesteps.

LocRNN's hidden state is composed of two neural populations, $\mathbf{L_t}$ and $\mathbf{S_t}$ at timestep $t$ respectively. $\mathbf{L_t}$ receives longer-range lateral connections, i.e., presynaptic neurons connecting to $L_t$ cover a wider range of image space than those connecting to $\mathbf{S_t}$. These two populations are analogous to the $x$ (excitatory) and $y$ (inhibitory) neurons in Li (1998). However, our implementation to retain this strict excitatory/inhibitory profile by restricting their recurrent weights to be strictly positive/negative did not converge in a stable manner. On the other hand, retention of two populations of neurons performed better than replacing them with a single uniform population. Initially, both $\mathbf{L_0}$ and $\mathbf{S_0}$ populations are set to a tensor of zeros with the same shape as the input $\mathbf{X}$.
$L$ and $S$ update gates $\mathbf{G_t^L}$ and $\mathbf{G_t^S}$ (same shape as $\mathbf{L}$ and $\mathbf{S}$) are computed as functions of the input and current hidden states $\mathbf{L_{t-1}}$ and $\mathbf{S_{t-1}}$ using 1x1 convolutions $\mathbf{U_{**}}$.

$$\mathbf{G_t^L} = \sigma(LN(\mathbf{U_{lx}} * \mathbf{X}) + LN(\mathbf{U_{ll}} * \mathbf{L_{t-1}})) \qquad (1)$$

$$\mathbf{G_t^S} = \sigma(LN(\mathbf{U_{sx}} * \mathbf{X}) + LN(\mathbf{U_{ss}} * \mathbf{S_{t-1}})) \qquad (2)$$

Each of the 4 types of lateral connections namely 1) $L \rightarrow L$, 2) $L \rightarrow S$, 3) $S \rightarrow S$, 4) $S \rightarrow L$ neurons are modeled by convolution kernels $\mathbf{W_{ll}}, \mathbf{W_{sl}}, \mathbf{W_{ss}}$, and $\mathbf{W_{ls}}$ respectively. $\mathbf{W_{ll}}$ and $\mathbf{W_{ls}}$ modeling long-range lateral interactions are of shape $d \times d \times h_l \times w_l$ where $d$ is the dimensionality of the hidden state and $h_l$ and $w_l$ represent the kernel spatial size. To be precise, each neuron at location $(i, j, m)$ in $\mathbf{L}$ (spatial location i,j and channel m) receives lateral connections from a window of neurons at locations $(x, y, n), x \in [i - h_l/2, i + h_l/2], y \in [j - w_l/2, j + w_l/2], n \in [0, k)$ in $\mathbf{L}$

and $\mathbf{S}$ mediated by $\mathbf{W_{ll}}$ and $\mathbf{W_{ls}}$ respectively. Similarly, $\mathbf{W_{ss}}, \mathbf{W_{sl}}$ are of shape $h_s$ and $w_s$ such that $(h_l, w_l) > (h_s, w_s)$.

$$\tilde{\mathbf{L}}_\mathbf{t} = \gamma(\mathbf{W_{lx}} * \mathbf{X} + \mathbf{W_{ll}} * \mathbf{L_{t-1}} + \mathbf{W_{ls}} * \mathbf{S_{t-1}}) \tag{3}$$

$$\tilde{\mathbf{S}}_\mathbf{t} = \gamma(\mathbf{W_{sx}} * \mathbf{X} + \mathbf{W_{sl}} * \mathbf{L_{t-1}} + \mathbf{W_{ss}} * \mathbf{S_{t-1}}) \tag{4}$$

Once the long-range lateral influences are computed and stored in $\tilde{\mathbf{L}}_\mathbf{t}$ and $\tilde{\mathbf{S}}_\mathbf{t}$, these are mixed with the previous hidden states using the gates computed in Eq. 1. These hidden states are then passed on to subsequent recurrent iterations where even longer-range interactions occur (as time increases).

$$\mathbf{L_t} = \kappa(LN(\mathbf{G_t^L} \odot \tilde{\mathbf{L}}_\mathbf{t} + (1 - \mathbf{G_t^L}) \odot \mathbf{L_{t-1}})) \tag{5}$$

$$\mathbf{S_t} = \kappa(LN(\mathbf{G_t^S} \odot \tilde{\mathbf{S}}_\mathbf{t} + (1 - \mathbf{G_t^S}) \odot \mathbf{S_{t-1}})) \tag{6}$$

In the above equations, $LN()$ stands for Layer Normalization (Ba et al., 2016), and the nonlinearities $\gamma$ and $\kappa$ are both set to ReLU.

## 4.2 MODEL FAMILIES EVALUATED ON TASK EXTRAPOLATION

As we have noted in Related Work, prior work has examined the relationship between recurrent networks, standard ResNets, and ResNets with weight sharing. Owing to commonalities between these classes of architectures, they are most relevant to studying task extrapolation; we design ResNets, Recurrent ResNets which we refer to as R-ResNets, and recurrent networks of varying effective depths. We describe details of our specific implementation (Anonymized code available at https://github.com/task-extrapolation/task-extrapolation.) of these architectures in this section.

**Specialized recurrent networks:** We evaluated the task extrapolation ability of the following specialized recurrent architectures. We evaluated the LocRNN model we described in detail in Sec. 4.1. In addition, we also evaluated the hConvGRU (Linsley et al., 2018), ConvRNN (Nayebi et al., 2022) and CORNet-S (Kubilius et al., 2018) all of which are relevant convolutional recurrent neural networks. The Gated Recurrent Unit is quite relevant to both these architectures and utilizes a similar gating strategy as LocRNN and hConvGRU. We evalute ConvGRU as well in the subsequent section on PathFinder, Mazes and on task extrapolation.

**Residual Networks (ResNets):** ResNets are not only high performing feature extractors for various downstream visual processing tasks, they also attain high scores on the Brain-Score leaderboard (Schrimpf & Kubilius, 2018) that measures representational similarity to biological vision. In this work we implement three ResNet architectures on the Mazes challenge and two ResNet architectures on the PathFinder challenge. On Mazes, we used 1) ResNet-12, 2) ResNet-20 and 3) ResNet-36 architectures while on PathFinder, we used 1) ResNet-10 and 2) ResNet-18 architectures; we implemented all these ResNet architectures to use the same width (# of output channels) at each residual block. This width parameter was set to 64 for ResNets evaluated on Mazes and to 54 for ResNets evaluated on PathFinder.

**Recurrent Residual Networks (R-ResNets):** We implemented R-ResNets by enabling weight-sharing across the stack of residual blocks and note that this is a much lighter / weaker notion of recurrence compared to specialized recurrent architectures. This is reflected in the raw performance difference and extrapolation performance difference in comparison to specialized recurrent architectures. R-ResNets are designed with an input convolution layer and output readout layers that together form 4 layers of processing. Each recurrent block that extracts features for PathFinder and Mazes is also 4 layers deep. Hence, the total number of layers in R-ResNets is a multiple of 4. To enable a fair comparison among recurrent networks we used the same number of recurrent iterations, kernel size and roughly matched the parameter count for all three recurrent networks on both tasks separately.

## 5 RESULTS

### 5.1 LOCRNN EXCELS BOTH AT MAZES AND PATHFINDER

Before testing for length generalization of our selected models (discussed in Sec. 4), we first evaluated models' raw ability to integrate long-range spatial dependencies on Pathfinder and Maze challenges. We systematically tested this ability by varying the difficulty of Pathfinder across 3 levels

parameterized by path length (9, 14, and 18), and the difficulty of Mazes across 3 levels of difficulty parameterized by the maze's grid size (9×9, 11×11, and 13×13 respectively). For each of the 6 datasets (3 Pathfinder and 3 Maze respectively), we used non-overlapping training and testing splits of the full datasets to train models and evaluate their performance during test-time.

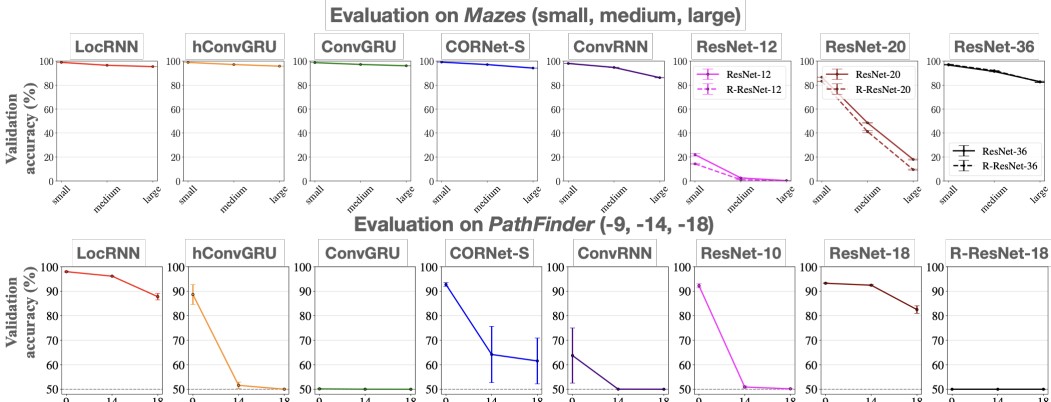

Figure 1: (Top) Model performance comparison on Mazes. All recurrent networks outperform the ResNet-based feedforward baselines. We observe a trend that increasing depth improves the performance of ResNets on Mazes. (Bottom) Model performance comparison on PathFinder. Differences between the recurrent networks becomes clearer on the more difficult task of PathFinder.

We created non-overlapping training and testing sets of Pathfinder and Maze datasets that were used to train the models and evaluate their test accuracy. All models used the same training and testing image sets.

Results from Fig. 1 (Top) shows that all recurrent networks outperform their ResNet-based feedforward baselines on Mazes (including the deepest ResNet-36 network) despite using many fewer free parameters. We replicate the trend observed by Schwarzschild et al. (2022): ResNets and R-ResNets (their recurrent counterparts) at matched effective depths are quite similar in their performance on the 3 Maze challenges. However we show that both ResNets and **R-ResNets are considerably different from all specialized recurrent architectures** (LocRNN, hConvGRU, ConvGRU) which are all more accurate and sample efficient compared to the ResNets. We would like to emphasize that this difference is significant and argues for improved effectiveness of specialized recurrent architectures in representing and reasoning with spatial context via integration of long-range spatial dependencies. On Mazes, we find that **by a narrow margin of 1.34%, ConvGRU outperforms LocRNN and is the best performing model on Mazes.**

On **PathFinder challenge, LocRNN is the best performing architecture across all difficulty levels** of the challenge including the most difficult PathFinder-18. Results on PathFinder show key differences between the recurrent architectures. Simple weight-tying based R-ResNet-18 is unable to learn even the easiest version of the challenge. On the other hand there exist significant differences between the sophisticated recurrent architectures LocRNN, ConvGRU, hConvGRU, CORNet-S (Kubilius et al., 2018) and ConvRNN (Nayebi et al., 2018). hConvGRU learns the easy version of PathFinder (PathFinder-9) but fails to converge on PathFinder-14 and PathFinder-18 [1]. Overall in this evaluation the LocRNN model shines with top performance on all 3 difficulty levels of both challenges either closely matching or outperforming feedforward ResNet- baselines with orders of magnitude more parameters. LocRNN's performance is followed by ResNet-18 and then CORNet-S which achieves better than chance performance (yet much inferior to LocRNN) on PathFinder-14 and PathFinder-18. Importantly, **ConvGRU and hConvGRU, the only models to match/outperform LocRNN on Mazes are significantly worse than LocRNN on PathFinder with ConvGRU being** *at chance* on all difficulty levels and hConvGRU being at chance on all but the easiest difficulty level.

---

[1] Training hConvGRU using code made available by Linsley et al. (2018) (using larger recurrent convolutional kernels and an order more GFLOPS than what we used here) in an architecture significantly heavier in GPU costs was able to solve PathFinder-14. However this training setting is too expensive and and we report results on modest hyperparameter settings on lightweight architectures used by all models.

## 5.2 RECURRENT NETWORKS EXHIBIT TASK EXTRAPOLATION

While models are trained on one difficulty level of a task, we define task extrapolation as the ability to dynamically reduce (and increase) computation to generalize zero-shot to easier (and harder) difficulty levels of the task without extra training data; this is impossible to achieve with feedforward networks aren't intrinsically suited to dynamically vary their computational budget during inference (for e.g. see recurrent segmentation at variable computational budgets (McIntosh et al., 2016)). However, as soon as feedforward networks use weight-tying, we refer to them as recurrent networks unrolling their temporal computation via depth and can hence equip variable computational budget during inference. We test the ability of various recurrent architectures from Sec. 5.1 in task extrapolation by altering their recurrent timesteps during inference.

For instance, the accuracy of LocRNN tested on Small Mazes while it was trained on Medium Mazes measures the extrapolation of LocRNN to an easier Maze task (Medium → Small). Although it may seem trivial for biological intelligence to solve easier problems after learning a hard problem, it need not necessarily be the case for neural networks which have no exposure to instances from an easier challenge. Hence we evaluate generalization to both easier and harder problems.

We chose those recurrent architectures with high performance on the medium Mazes and PathFinder challenges for their respective task extrapolation experiments. Remaining models with poor or chance performance on these tasks are not fit for the evaluation of extrapolation as their within-distribution performance is itself lacking. We reused the following architectures based on the above criteria for evaluating task extrapolation – a) LocRNN, b) ConvGRU, and c) R-ResNet-36 trained on Medium Mazes and only LocRNN trained on PathFinder-14 (since no other recurrent network was able to converge to accurate solutions on PathFinder-14). Per-timestep batch normalization layers in hConvGRU prevent us from performing inference with this model at arbitrary number of iterations aside from what was used during training, hence we could not evaluate its task extrapolation performance other than with the same number of steps it trained on (this is discussed below). We evaluated the zero-shot generalization performance of the above architectures on easier and harder variants of Mazes and PathFinder with $t < T$ (for easier) and $t > T$ (for harder) recurrent steps where $T$ denotes the number of recurrent steps used during training.

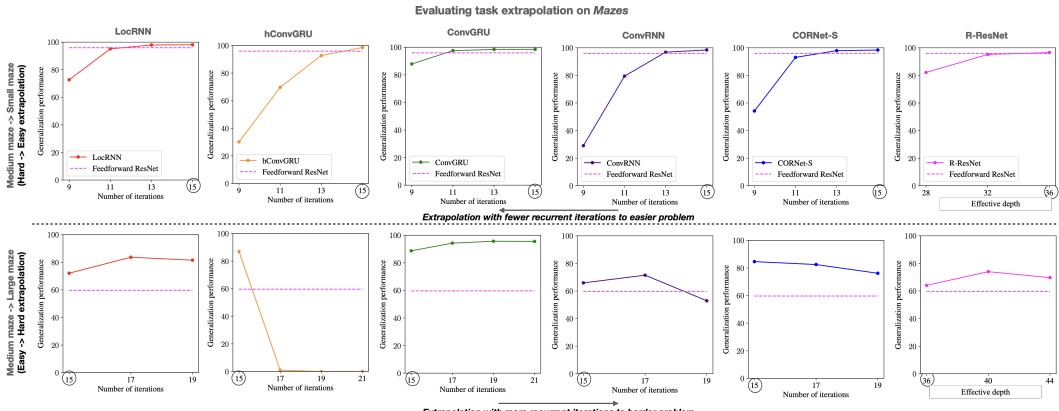

Figure 2: (Top) Task extrapolation performance of models to easier (Small) maze by decreasing number of recurrent steps as we move horizontally in each plot from the right (timesteps used during training) to the left. (Bottom) Task extrapolation performance of models to harder (Large) maze by increasing number of recurrent steps as we move horizontally in each plot from the left (timesteps used during training) to the right. Dotted blue line shows the zero-shot generalization performance of the feedforward ResNet-36 model to small and large mazes. ResNet-36 is significantly worse than all recurrent models in generalizing to the harder Maze. Number of recurrent timesteps used during training is circled on each plot's x-axis.

**All RNNs adapt recurrence to show extrapolation** Each plot in Fig. 2.A shows the extrapolation performance of different models on Mazes. From these plots we see that recurrent networks in general are able to gain more performance on easier and harder mazes simply by using fewer and more recurrent steps than what they were trained with (highlighted with circles on the x-axis).

**Specialized recurrent architectures show stronger extrapolation:** As we've noted before, R-ResNet uses a weak form of recurrence by using weight-tying across layers of a feedforward network, this difference reflects on the model's inferior generalization performance relative to specialized recurrent architectures we evaluated. Fig. 2 shows that ConvGRU designed particularly for sequential processing is overall the best-performing model (95.6%) on extrapolating to large mazes. On small mazes, all recurrent architectures have matching extrapolation performance and are still better than R-ResNet. The architectures CORNet-S and hConvGRU use learnable BatchNorm parameters at each timestep, preventing us from running them for more than the number of timesteps they were trained with. However, we use the following workaround: we perform a soft restoration of learned weights; up until the number of timesteps used during training the per-timestep learned BatchNorm parameters are restored from the trained model. During inference on greater number of timesteps than what was used during training, we initialize these learnable BatchNorm parameters at random.

**LocRNN shows strong extrapolation on the harder PathFinder challenge** We show extrapolation to easier (PathFinder-9) and harder (PathFinder-18) PathFinder problems in Fig. 3.A. LocRNN shows strong extrapolation performance to the easier and harder PathFinder challenge images. While generalizing to PathFinder-9, LocRNN grows from chance performance at 12 steps of recurrence (used during training) to achieving $> 90\%$ accuracy on a reduced computational budget (8 steps). On the harder PathFinder-18 problem, while it is at chance when run for 12 recurrent steps (used during training), increasing recurrence to 14 steps allows the model to reach a nontrivial accuracy of $62\%$ matching the much more parameter heavy feedforward ResNet's generalization performance.

### 5.3 ABLATION STUDY OF LOCRNN

**Interneurons are crucial for good performance:** We found strong evidence suggesting that interneurons are crucial to LocRNN's high performance. First, we retained L neurons and removed S neurons (which are interneurons in LocRNN as mentioned above) in LocRNN (-S neurons in Fig.3B). This ablation converges to a stable solution in only 1 out of 3 random seeds (also with a 38% drop in sample efficiency). Next, to match the number of parameters, we increased the width of L neurons to 64 channels (-S neurons + 2x L neurons in Fig.3B). While there was a slight improvement (2 out of 3 seeds converged), the sample efficiency of the models that converged was still very low compared to the full model (25% drop in sample efficiency). In an interesting third ablation experiment, we **retained both L and S neurons and sent both neuron populations upward** for classification. This explicitly tests whether the interneuron property helps improve expressivity. Again, only one of the 3 seeds converged to an accurate solution and with very poor sample efficiency, highlighting that the property of S neurons being **interneurons is of utmost importance for LocRNN's high performance**.

**ReLU is important for good performance:** Having a long range of linear (non-saturating) operating mode in the activation function is critical to LocRNN's performance. Replacing ReLU with tanh completely breaks LocRNN's performance. However, replacing ReLU with a clipped ReLU (clipped at 6) did not cause any drop in performance. Also, in agreement with this, when we replace tanh in ConvGRU with ReLU, we achieve a significant gain in ConvGRU's PathFinder-14 performance (+10.4%).

**Gating and LayerNorm contribute to improving performance and sample efficiency:** We find that LayerNorm affects performance the least among all components. Even in the absence of LN, LocRNN achieves $> 80\%$ performance on PathFinder. However, LN seems to improve sample efficiency significantly, similar to gating; the absence of which hurts both performance and sample efficiency.

### 5.4 A TRADEOFF BETWEEN TASK-OPTIMIZATION AND EXTRAPOLATION

Our observation comparing the performance of recurrent architectures in Fig. 1 and Fig. 2.A highlight an interesting conundrum. While LocRNN is the only recurrent architecture to achieve near-perfect accuracy on both Mazes and PathFinder, ConvGRU (which is much inferior to LocRNN on the PathFinder challenge) produces the best extrapolation performance on Mazes. This suggests an intriguing tradeoff between task performance optimization and stability in hidden state space. ConvGRU is inductively biased by design to be more stable due to the tanh nonlinearity used in

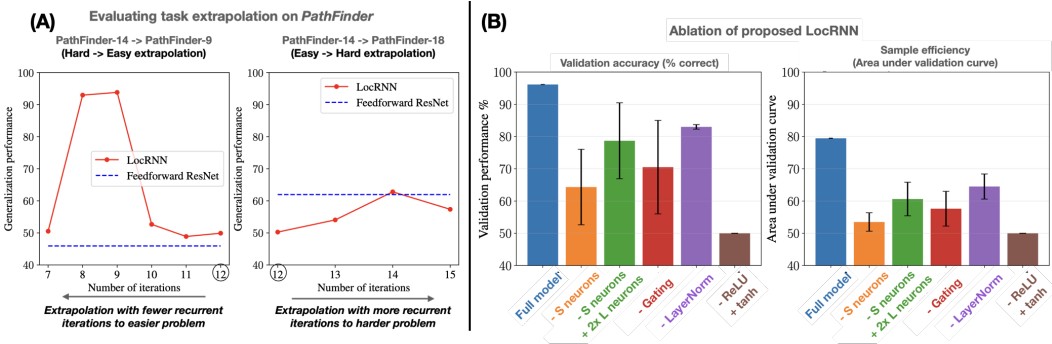

Figure 3: (A) Task extrapolation performance of LocRNN from Pathfinder-14 to Pathfinder-9 and PathFinder-18. ConvGRU and R-ResNet not shown here as they were unable to converge on PathFinder within the provided training budget. The number of recurrent steps used during training is highlighted with a circle on the x-axis of each plot. (B) Validation accuracy and sample efficiency of various ablations of LocRNN.

its output layer (a saturating nonlinearity that has bounded outputs) whereas LocRNN uses a ReLU nonlinearity with unbounded outputs. This could be the underlying reason why LocRNN is unstable yet a high performing model. In fact, replacing the $\tanh$ in ConvGRU with ReLU enhances the model's performance from chance to 90.05% and 60% on PathFinder-9 and PathFinder-14 respectively.

The high task performance of LocRNN potentially comes at the cost of reduced stability in the hidden state space with non-smooth dynamics around the number of training recurrent steps. This is evident from our visualization of the validation accuracy of LocRNN and ConvGRU as a function of the number of recurrent steps in Fig. 7 (in Appendix). Our future work explores techniques to improve the stability of LocRNN in order to unlock a class of recurrent neural networks that can be optimized on any given task to produce high performance as well as learn stable state space representations as a function of recurrent iterations.

# 6 DISCUSSION

Extrapolating to new easier and harder instantiations of a task is a kind of out-of-distribution generalization where agents learn to dynamically adapt their learned algorithms to solve new easier / harder versions of a problem. Ullman (1984) visual routines provide a conceptual framework to accomplish the development and deployment of such flexible algorithms based on the recurrent (sequential) application of a set of elemental operations within human cognition. We combine this conceptual framework with Li (1998)'s computational model of recurrence in primate early vision and a state of the art recurrent neural architecture to demonstrate that neural networks can also exhibit task extrapolation which is a trademark characteristic of human cognition. As part of this work we introduced a high-performing recurrent architecture called LocRNN that achieves excellent accuracy and sample efficiency on two challenging visual spatial reasoning benchmarks, Mazes and PathFinder (best performing model on this task). An ablation study of LocRNN reveals the individual contributions of each of its components and provides a mechanistic understanding of LocRNN's high performance. LocRNN and other recurrent architectures exhibited task extrapolation on both the above tasks in contrast to feedforward networks that are restricted by their fixed computational budget during inference which cannot accomplish such flexible generalization. We believe that the abundance of recurrent processing in biological neurons underlies human ability to exhibit task extrapolation. Upon comparing the performance of LocRNN with other recurrent architectures, we develop the key insight of a potential trade-off between stability and task performance in recurrent architectures. Our future work will examine the development of a new class of stable recurrent architectures that, while using unbounded activation functions such as ReLU (that contribute high expressivity), are also jointly designed/optimized to produce stable recurrent activations which show strong task extrapolation behavior. Our work argues that recurrent operations are fundamental to biological and artificial vision and inspires future work studying the utility of recurrent architectures in the performance and out-of-distribution generalizability of deep learning models.

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

## A  APPENDIX

### A.1  TRAINING DETAILS

All experiments were performed using PyTorch (Paszke et al., 2017). On Mazes, we trained models using per-pixel binary cross entropy with a minibatch size of 256 images and a learning rate schedule starting with warmup followed by learning rate decay as indicated in Schwarzschild et al. (2021) for 50 total epochs of training. On PathFinder-14, we used binary cross-entropy to train models with a minibatch size of 256 images and a constant learning rate of 1e-4 for all models for a total of 5 epochs of training (**LocRNN often learns to solve the task in less than 1 epoch and is the most sample efficient model**). For models that did not converge, we varied the learning rate in steps of 0.1 lesser and greater than 1e-4 but still confirm that they do not converge. Models are quite robust to the choice of learning rate on both PathFinder and Mazes.

### A.2  PER-TIMESTEP PREDICTIONS FROM LOCRNN ON MEDIUM MAZES

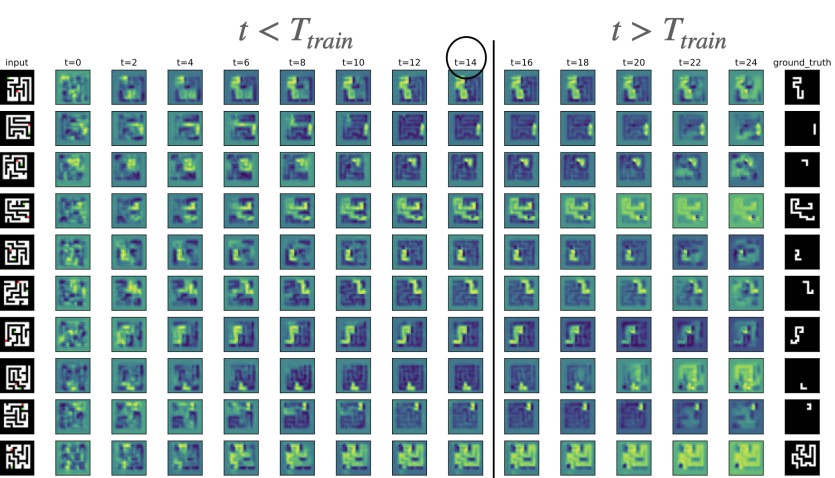

Figure 4: Visualizing predictions of LocRNN (trained on medium mazes) for Medium Maze image samples at subsequent timesteps shows that the model is most stable near the number of recurrent steps used during training. As we increase $t > T$ used during train, the predictions of maze solution become blurred and the overall activity of the prediction grows without bounds.

### A.3  EXAMPLE IMAGES FROM PATHFINDER AND MAZES

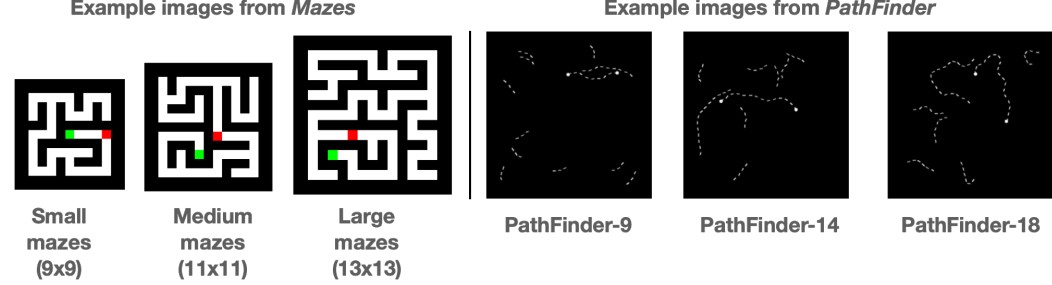

Figure 5: (Left) Example images from Mazes at three difficult levels (small, medium, large); (Right) Example images from PathFinder at three difficulty levels (PathFinder-9, PathFinder-14, PathFinder-18). Scales for different difficulties within Mazes are as shown; Pathfinder images have five times the resolution of Large Mazes.

## A.4 INPUT AND OUTPUT FORMAT FOR PATHFINDER AND MAZES

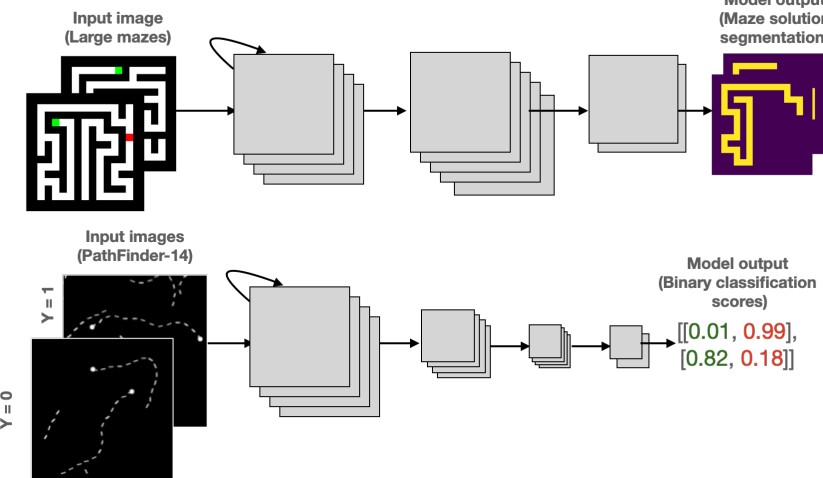

Figure 6: (Top) Example images from Large Mazes processed by a model to produce the solution as a segmentation prediction. (Bottom) Example input images from PathFinder-14 processed by a classifier to produce binary classification output.

## A.5 MODEL STABILITY ON MEDIUM MAZES

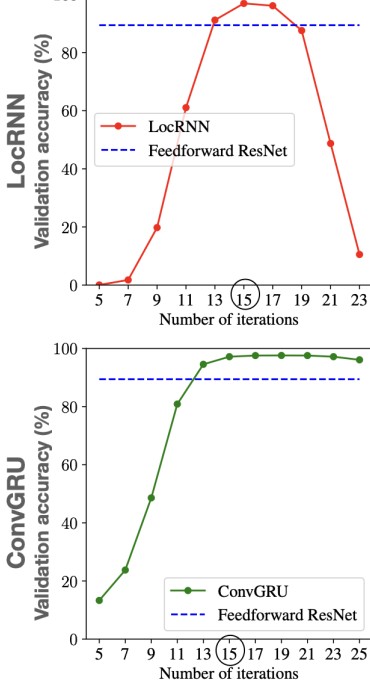

Figure 7: Visualizing validation accuracy of LocRNN (top) and ConvGRU (bottom) as a function of the recurrent iterations. Number of iterations used during training is highlighted with a circle on the horizontal axis. We observe that LocRNN is unstable when we heavily increase the number of recurrent iterations in contrast to the more stable ConvGRU.

## A.6 MODEL ARCHITECTURE DETAILS (LAYER WISE CONFIGURATION)

| Layer | Layer type | Kernel size | N-out | Pooling | Input | Output |
|---|---|---|---|---|---|---|
| LocRNN-Maze | | | | | | |
| 1 | Conv2D | 3*3 | 64 | None | [H,W,3] | [H,W,64] |
| 2 | DaleRNNLayer | divnorm 3*3 Exc 5*5 Inh 3*3 | 64 | None | [H,W,64] | [H,W,64] |
| 3 | Conv2D | 3*3 | 32 | None | [H,W,64] | [H,W,32] |
| 4 | Conv2D | 3*3 | 8 | None | [H,W,32] | [H,W,8] |
| 5 | Conv2D | 3*3 | 2 | None | [H,W,8] | [H,W,2] |
| hGRU-Maze | | | | | | |
| 1 | Conv2D | 3*3 | 64 | None | [H,W,3] | [H,W,64] |
| 2 | hConvGRU | 5*5 | 64 | None | [H,W,64] | [H,W,64] |
| 3 | Conv2D | 3*3 | 32 | None | [H,W,64] | [H,W,32] |
| 4 | Conv2D | 3*3 | 8 | None | [H,W,32] | [H,W,8] |
| 5 | Conv2D | 3*3 | 2 | None | [H,W,8] | [H,W,2] |
| GRU-Maze | | | | | | |
| 1 | Conv2D | 3*3 | 64 | None | [H,W,3] | [H,W,64] |
| 2 | ConvGRU | 5*5 | 64 | None | [H,W,64] | [H,W,64] |
| 3 | Conv2D | 3*3 | 32 | None | [H,W,64] | [H,W,32] |
| 4 | Conv2D | 3*3 | 8 | None | [H,W,32] | [H,W,8] |
| 5 | Conv2D | 3*3 | 2 | None | [H,W,8] | [H,W,2] |

| Layer | Layer type | Kernel size | N-out | Pooling | Input | Output |
|---|---|---|---|---|---|---|
| ResNet-12-Maze | | | | | | |
| 1 | Conv2D | 3*3 | 64 | None | [H,W,3] | [H,W,64] |
| 4*Block | Conv2D | 3*3 | 64 | None | [H,W,64] | [H,W,64] |
| | Conv2D | 3*3 | 64 | None | [H,W,64] | [H,W,64] |
| | shortcut | | | | | |
| 3 | Conv2D | 3*3 | 32 | None | [H,W,64] | [H,W,32] |
| 4 | Conv2D | 3*3 | 8 | None | [H,W,32] | [H,W,8] |
| 5 | Conv2D | 3*3 | 2 | None | [H,W,8] | [H,W,2] |
| ResNet-20-Maze | | | | | | |
| 1 | Conv2D | 3*3 | 64 | None | [H,W,3] | [H,W,64] |
| 8*Block | Conv2D | 3*3 | 64 | None | [H,W,64] | [H,W,64] |
| | Conv2D | 3*3 | 64 | None | [H,W,64] | [H,W,64] |
| | shortcut | | | | | |
| 3 | Conv2D | 3*3 | 32 | None | [H,W,64] | [H,W,32] |
| 4 | Conv2D | 3*3 | 8 | None | [H,W,32] | [H,W,8] |
| 5 | Conv2D | 3*3 | 2 | None | [H,W,8] | [H,W,2] |
| ResNet-36-Maze | | | | | | |
| 1 | Conv2D | 3*3 | 64 | None | [H,W,3] | [H,W,64] |
| 16*Block | Conv2D | 3*3 | 64 | None | [H,W,64] | [H,W,64] |
| | Conv2D | 3*3 | 64 | None | [H,W,64] | [H,W,64] |
| | shortcut | | | | | |
| 3 | Conv2D | 3*3 | 32 | None | [H,W,64] | [H,W,32] |
| 4 | Conv2D | 3*3 | 8 | None | [H,W,32] | [H,W,8] |
| 5 | Conv2D | 3*3 | 2 | None | [H,W,8] | [H,W,2] |

| Layer | Layer type | Kernel size | N-out | Pooling | Input | Output |
|---|---|---|---|---|---|---|
| R-ResNet-12-Maze | | | | | | |
| 1 | Conv2D | 3*3 | 64 | None | [H,W,3] | [H,W,64] |
| 4iter*Block | Conv2D | 3*3 | 64 | None | [H,W,64] | [H,W,64] |
| | Conv2D | 3*3 | 64 | None | [H,W,64] | [H,W,64] |
| | shortcut | | | | | |
| 3 | Conv2D | 3*3 | 32 | None | [H,W,64] | [H,W,32] |
| 4 | Conv2D | 3*3 | 8 | None | [H,W,32] | [H,W,8] |
| 5 | Conv2D | 3*3 | 2 | None | [H,W,8] | [H,W,2] |
| R-ResNet-20-Maze | | | | | | |
| 1 | Conv2D | 3*3 | 64 | None | [H,W,3] | [H,W,64] |
| 8iter*Block | Conv2D | 3*3 | 64 | None | [H,W,64] | [H,W,64] |
| | Conv2D | 3*3 | 64 | None | [H,W,64] | [H,W,64] |
| | shortcut | | | | | |
| 3 | Conv2D | 3*3 | 32 | None | [H,W,64] | [H,W,32] |
| 4 | Conv2D | 3*3 | 8 | None | [H,W,32] | [H,W,8] |
| 5 | Conv2D | 3*3 | 2 | None | [H,W,8] | [H,W,2] |
| R-ResNet-36-Maze | | | | | | |
| 1 | Conv2D | 3*3 | 64 | None | [H,W,3] | [H,W,64] |
| 16iter*Block | Conv2D | 3*3 | 64 | None | [H,W,64] | [H,W,64] |
| | Conv2D | 3*3 | 64 | None | [H,W,64] | [H,W,64] |
| | shortcut | | | | | |
| 3 | Conv2D | 3*3 | 32 | None | [H,W,64] | [H,W,32] |
| 4 | Conv2D | 3*3 | 8 | None | [H,W,32] | [H,W,8] |
| 5 | Conv2D | 3*3 | 2 | None | [H,W,8] | [H,W,2] |

| Layer | Layer type | Kernel size | N-out | Pooling | Input | Output |
|---|---|---|---|---|---|---|
| LocRNN-Pathfinder | | | | | | |
| 1 | Gabor Filter | 3*3 | 32 | Max | [H,W,3] | [H/4,W/4,32] |
| 2 | DaleRNNLayer | divnorm 5*5 Exc 9*9 Inh 5*5 | 32 | None | [H/4,W/4,32] | [H/4,W/4,32] |
| 3 | Conv2D | 1*1 | 2 | Max | [H/4,W/4,32] | [2] |
| 4 | Linear | - | 2 | None | [2] | [2] |
| hGRU-Pathfinder | | | | | | |
| 1 | Gabor Filter | 3*3 | 32 | Max | [H,W,3] | [H/4,W/4,32] |
| 2 | hConvGRU | 9*9 | 38 | None | [H/4,W/4,32] | [H/4,W/4,32] |
| 3 | Conv2D | 1*1 | 2 | Max | [H/4,W/4,32] | [2] |
| 4 | Linear | - | 2 | None | [2] | [2] |
| GRU-Pathfinder | | | | | | |
| 1 | Gabor Filter | 3*3 | 32 | Max | [H,W,3] | [H/4,W/4,32] |
| 2 | ConvGRU | 9*9 | 24 | None | [H/4,W/4,32] | [H/4,W/4,32] |
| 3 | Conv2D | 1*1 | 2 | Max | [H/4,W/4,32] | [2] |
| 4 | Linear | - | 2 | None | [2] | [2] |

