# OpenReview forum: "Cortically motivated recurrence enables task extrapolation"
_ICLR.cc/2023/Conference — Submitted to ICLR 2023_

### Official Review · Reviewer_3CSc · 2022-10-19

**Confidence:** 4
**Correctness:** 3
**Technical Novelty And Significance:** 4
**Empirical Novelty And Significance:** 4
**Recommendation:** 6

**Clarity, Quality, Novelty And Reproducibility:**

The writing is clear, except for a couple minor points above. The overall flow of the paper is sensible, and the the choice of datasets as well as models to compare with is good. The quality is high -- the experimental results support the main claims. As far as I know, the work is an original contribution building on other important works in the field and prompting even more research into recurrence and task extrapolation.

**Strength And Weaknesses:**

Strengths:
- The paper flows well.
- The experiments are thorough and illuminating.
- The writing is mostly clear.
- The architecture is meaningfully different from existing techniques, and shows a real benefit over those methods.

Weaknesses:
- I don’t understand the point about convergence and computational budget. Are there models that didn't train successfully? What was tried here? More specifically, in the footnote, it sounds like these models that don't train are paired down, lighter weight versions than were originally proposed -- do the full-strength models train? How is pairing down the model size (and computation budget) justified if these networks don't train?
- The results that aim to show that LocRNN is better on mazes is not so compelling. In Figures 2 and 3 ConvGRU looks superior on Mazes. The text and the captions seem to minimize this, so at best the difference is unclear to the reader. More information and clearer claims around this are needed.
- Code is missing. I followed the link to the anonymized GitHub repository on October 17th and it was empty. Particularly, when it is stated that code is available, I take issue with this. I understand the criteria for submission do not require code, but I don't think there is enough information to reliably reproduce the results here *and* the authors state that code is available. This needs to be addressed.

Two minor issues (not affecting score):
- There is a bad reference to a "Fig 5.2" in the second paragraph on page 8.
- The second sentence of section 5.3 is unclear: “LocRNN is the best performing architecture that outperforms all
compared baselines in terms of raw performance, whereas ConvGRU and hConvGRU which are much inferior to LocRNN on the PathFinder challenge reach top extrapolation performance.”


**Summary Of The Paper:**

A novel recurrent architecture is introduced with the express goal of learning models that can generalize to examples that are more/less difficult than the training data. On two benchmark datasets with test sets of several difficulties, this new architecture beats existing models on out of distribution performance.

**Summary Of The Review:**

I find the paper solid. There are a couple weaknesses and a couple minor points that can be addressed to improve this paper. I look forward to the authors' response. Should my concerns be adequately addressed, I'll be happy to raise my score.

---

> ### Author Response · Authors · 2022-11-19
> **Authors' response to comments from Reviewer 3CSc**
>
> Dear reviewer,
>
> We thank you for your time and careful consideration of our submission. **We have uploaded a revised version of our submission during the revision phase with several new experiments and findings addressing reviewer comments.** We are highly encouraged to know your positive feedback on the paper and your constructive comments to improve our submission. In line with the reviewers' feedback on experimentation, we have added several interesting experiments and an ablation study of LocRNN’s components to better interpret the algorithmic underpinning of each of these parts of LocRNN. We request you to kindly refer to our common response where we discuss our observations on these above-mentioned experiments using more convolutional recurrent baselines (Nayebi et al 2021, Kubilius et al 2018), ablation, and a note on the relevance of our evaluated tasks to biological vision and the difficulty of performing these tasks. Following is our response to your unique comments on our submission:
>
> - **On the convergence of hGRU:** “Are there models that didn’t train successfully?” – Yes, there were several models that didn’t train successfully in the most difficult settings (PathFinder-14, -18) as highlighted by their chance performance on Fig. 1. Our goal is to use a common base architecture and vary the intermediate processing to be one of the different baseline models we evaluate. We standardize all factors of architectural variability to our knowledge (such as input convolution layers, readout, preprocessing, augmentation etc.) other than the individual model components to purely estimate model abilities under modest training costs taking into account the fact that we need to train 3 random initializations of ~10 baselines on 1,000,000 image (PathFinder) and 60,000 image (Mazes) datasets.
>     *On the footnote that was specific to hConvGRU:*
> We would like to note that all our models use the same base architecture. We choose this architecture based on prior experiments using the kind of datasets we work with and our GPU compute setup’s bottlenecks. **In this common and fair setting that we use to evaluate all models, hGRU fails to converge on PathFinder-14 and PathFinder-18**. However, there is a particular compute heavy hConvGRU architecture that Linsley et al release as part of their code. **While LocRNN uses 440 GFLOPs at a minibatch size of 256 images, the code released by Linsley et al uses an enormous 6524.35 GFLOPs of compute at a minibatch size of 128 images!!** This compute-intensive version of base architecture and hConvGRU *is highly prohibitive for training on our compute resources and deviates from our goal*. We are NOT interested in scaling up models to use large number of parameters/ GFLOPs and seeing if they learn PathFinder or Mazes in the limit. On the contrary, we are interested in using a lightweight base architecture that trains quickly and seeing which of the baselines learn these tasks and extrapolate. Still, we trained one hConvGRU in this high-compute setting and it did converge, however, the training took more than a day to complete whereas LocRNN and hConvGRU in our base architecture trains in less than a few hours.
> - **Re performance of LocRNN on Mazes:** Thank you for highlighting the lack of clarity in our explanation of LocRNN’s performance on Mazes. *As you rightly pointed out, ConvGRU outperformed LocRNN on Mazes* and we emphasize this point more clearly in our updated revision. **However, as we also mentioned in our response to i8Ub, we would like the readers and reviewers to please take this result in conjunction with ConvGRU’s performance at chance on PathFinder.** While most models do well on Mazes, only LocRNN and ResNet-18 (10x more parameters than LocRNN) are capable of accurately solving PathFinder-14 and PathFinder-18.
> - **Re second sentence of Section 5.3 on comparing LocRNN, ConvGRU and hConvGRU:** We have clarified our statement and added more detail to convey our message about comparing these three models.
> - **Code had been pushed on Oct 17, 2022:** We really appreciate your interest in our code release. We deeply care about fellow researchers being able to reproduce our empirical findings from this large-scale recurrence + extrapolation study.
> We released the code on the date of October 17, 2022, but it is likely that the reviewer might have checked a few hours before we made the upload.
> We apologize for this inconvenience, we realize that we must have uploaded the code at the time reviewing began to allow for reviewers to view our code but the extra time we took was used to ensure complete anonymity and not reveal any of the authors’ identities or affiliations.
> - **Bad reference to Fig 5.2:** Bad reference to a Fig. 5.2 has been fixed in the paragraph discussing LocRNN’s generalization performance on PathFinder, thank you very much for catching and highlighting this reference error.

---

> > ### Comment · Reviewer_3CSc · 2022-11-28
> > **Reviewer follow up**
> >
> > I appreciate the time and energy the authors put into their response. My concerns are addressed and I vote to accept this paper.

---

> > > ### Author Response · Authors · 2022-11-29
> > > **Thank you for acknowledging our response**
> > >
> > > Dear Reviewer,
> > >
> > > We thank you for your efforts and helpful comments on our paper. We are glad that your concerns were addressed by our response and that you vote to accept the paper.
> > >
> > > Best,
> > >
> > > Paper3171 Authors

---

### Official Review · Reviewer_vnpB · 2022-10-24

**Confidence:** 3
**Correctness:** 4
**Technical Novelty And Significance:** 3
**Empirical Novelty And Significance:** 2
**Recommendation:** 6

**Clarity, Quality, Novelty And Reproducibility:**

**Quality**
The proposed method is not theoretically motivated, but this is acceptable if there strong empirical results, particularly since the method is biologically motivated. However, there are unfortunately some weaknesses with the empirical results as noted above. Nevertheless, the proposed direction is interesting, and the paper could be a strong contribution if the authors can convingly show that LocRNN strongly outperforms baselines on challenging benchmarks.

**Clarity**
The paper is clear for the most part. The bolded sentences and paragraph headings in sections 4 and 5 are helpful. However, there are a few minor areas where the clarity could be improved:
1. It may help to illustrate the architecture of LocRNN (particularly the receptive fields).
2. It may be better to define LN right after eqns 1 and 2.
3. In the "LocRNN shows strong extrapolation on the harder PathFinder challenge" paragraph, Fig 5.2 doesn't appear to exist.

**Originality**
The proposed method appears novel; to my knowledge, there are no similar methods applied to visual task extrapolation.

**Strength And Weaknesses:**

**Strengths**
The authors investigate a potential solution to the problem of task extrapolation, which is an important area, and moreover take inspiration from research on visual routines. The proposed LocRNN method is well explained and has a good biological motivation. In the experiments, a number of good baselines are compared. Analysis of the experiments is strong; in particular, the tradeoff between extrapolation and stability is a nice addition to this section.

**Weaknesses**
In my view, the main weakness with this paper is with the empirical results. First, the Mazes and Pathfinder tasks may be a little too simple. If the goal of this paper is to propose an RNN solution to general visual task learning, then it would be better to experiment with more complex tasks. Otherwise, the significance of the paper is unfortunately limited.

Also, it is unclear how much better LocRNN performs compared to the baselines. For example, in Figure 2 it appears that ResNet-18 performs similarly to LocRNN on Pathfinder. Also, as the authors point out, on Mazes, ConvGRU outperforms LocRNN on task extrapolation to large mazes. I may be missing a key result, but I unfortunately don't see where LocRNN significantly outperforms the baselines.

As a more minor point, I'm unsure in section 5.3 if the instability of LocRNN is what causes better performance. The authors propose a potential explanation for why this may be the case, but it would help to have an experiment justifying this explanation. For example, how would ConvGRU perform with ReLU activation?

Finally, it would be nice to include some ablation experiments on LocRNN. What aspect of LocRNN makes it effective compared to the baselines (e.g. is it the gating mechanism or the division into two neural populations)?

**Summary Of The Paper:**

The paper proposes a recurrent neural network architecture suited for task extrapolation, in which the level of computation required to solve a problem can be dynamically altered depending on the task's difficulty. The architecture is biologically-motivated, and the authors demonstrate performance on the Mazes and Pathfinder tasks. Experiments reveal that the proposed method, LocRNN, is successfully able to perform task extrapolation and outperforms other baselines. The authors also find an intriguing tradeoff between performance on task extrapolation and stability.

**Summary Of The Review:**

Overall, the paper proposes an interesting, biologically-motivated solution to the problem of visual task extrapolation. Unfortunately, the empirical results are a little weak which limits the significance of the paper in its current form. If the experiments can be improved, the paper would be a much stronger contribution.

---

> ### Author Response · Authors · 2022-11-19
> **Authors' response to comments form Reviewer vnpB**
>
> Dear reviewer,
>
> We thank you for your time and careful review of our submission. **We have uploaded a revised version of our submission during the revision phase with several new experiments and findings addressing reviewer comments.** Thank you for your interest in expanding the experiments in our paper. In line with your feedback and that of other reviewers on experimentation, we have added several interesting experiments (including the very interesting baseline you suggested of GRU + ReLU, thank you very much for this suggestion!) and an ablation study of LocRNN’s components. We request you to kindly refer to our common response where we discuss our observations on further experimentation using more convolutional recurrent baselines (Nayebi et al 2021, Kubilius et al 2018), ablation, and a note on the relevance of our evaluated tasks to biological vision and the difficulty of performing these tasks. Following is our response to your unique comments on our submission:
>
> - **Evaluating ConvGRU with ReLU:** We really appreciate your suggestion of evaluating ConvGRU with the ReLU nonlinearity, this is an excellent baseline to test whether the usage of ReLU as we hypothesized in our submission promotes task performance over tanh. We ran this experiment and found convincing results suggesting that our hypothesis holds: ConvGRU w/ the ReLU nonlinearity 90.05% on PathFinder-9 and 60% on PathFinder-14 which is significantly better than chance performance of ConvGRU w/ tanh. However, ConvGRU w/ ReLU still does not learn PathFinder-18 which is still solved only by LocRNN and PathFinder with >80% accuracy.
> - **LocRNN vs ResNet-18 on PathFinder**:
> LocRNN performance on PathFinder-9, -14, -18 = (97.99, 96.16, 87.79) whereas ResNet-18 accuracy on PathFinder-9, -14, -18 = (93.27, 92.47, 82.51). LocRNN outperforms ResNet-18 (10x more parameters than LocRNN) on all difficulty levels of PathFinder making LocRNN the best-performing model across all models evaluated on PathFinder.
> - **Re the tasks used for our evaluation:** As per your feedback and that of i8Ub, we will be performing experiments evaluating LocRNN’s performance on new tasks in future scope. However, we find these new experiments to be out of the scope of our current submission where our goal is to estimate the ability of deep learning models to integrate long-range spatial dependencies and to perform task extrapolation.
> Adding another task significantly increases our computation cost; we train between 6-9 baseline models (w/ 3 random seeds) on 4 kinds of experiments (evaluation on Mazes, PathFinder, task extrapolation on Mazes, PathFinder). Adding another task to this pipeline would mean 2 more additional kinds of experiments on all 6-9 baselines w/ 3 random seeds which can be highly expensive to run.
> As noted by Reviewer 3CSc, we believe the following statement they make is accurate “the choice of datasets as well as models to compare with is good.”. For the context of our questions addressed in this paper, we request you please reconsider the additional requirement of more tasks.

---

> > ### Comment · Reviewer_vnpB · 2022-11-30
> > **Thank you for your response!**
> >
> > I appreciate the authors' detailed response and additional experiments. This addresses most of my concerns. Thus, I have increased my recommendation.

---

### Official Review · Reviewer_j482 · 2022-10-25

**Confidence:** 4
**Correctness:** 3
**Technical Novelty And Significance:** 2
**Empirical Novelty And Significance:** 3
**Recommendation:** 3

**Clarity, Quality, Novelty And Reproducibility:**

The paper would benefit from a more in-depth presentation and analysis of the LocRNN module, and how it relates to other RNN modules and it's computational properties (stability, vanishing and exploding gradient, etc).

**Strength And Weaknesses:**

The paper has some very interesting elements, starting from the biological and fundamental science motivation. However, it still needs some work in terms of providing mechanistic and algorithmic understanding on how the cortically motivated recurrent motif accomplishes what it accomplishes, and establishing the relevance of the specific path integration tasks in the context of the cortical substrate that the paper purportedly examines. More in detail, here are some of the strengths and weaknesses of the paper.

Strengths:
- The paper is trying to connect known fact of the anatomy and functionalities of primary visual cortex of the brain to the engineering of deep learning. In particular, the fact that the connectivity of the cortex is highly recurrent and it seems to be organized in terms of clearly defined short- and long-range connections. This sort of investigation seems very promising as it may help elucidate the algorithmic properties of neuro-physiological and anatomical structures.
- The paper investigate relevant metrics beyond accuracy, such as robustness as a function of recurrent steps, and generalization across task difficulty.

Weaknesses:
- The paper is in part motivated by the statement that feedforward networks are "strictly restricted" and "cannot dynamically change their computational graph". On the other hand, conditional modules like conditional batchnorm or context-dependent representations like those provided by transformers strongly contradict this claim. In addition, a recurrent architecture when unfolded through time can be thought of as a feedforward network with tied weight (as the paper also recognizes when examining the R-ResNet architecture). It is then not clear what exactly this claim regarding the limitations of feedforward architectures alludes to exactly, since it would also pertain to RNNs (being a special case of feedforward architecture). It would be beneficial to re-target this claim and re-contextualize in light of these two objections.
- The paper proposes a new recurrent module without however providing any quantitative analysis of its functioning. It would be interesting if the paper would also include an analysis of the stability of module examining the vanishing and exploding gradient problems.
- The paper is highly focused on a very restricted and idiosyncratic type of task (path finding) which isn't  clearly motivated in terms of being of wide enough interest for the machine learning and deep learning communities at the moment. This could be mitigated by arguing that these tasks are at least a good fit for the cortical-type of architectures that are then proposed. In other words, what is the evidence that these tasks are paradigmatic of the type of tasks that the visual cortex and its architecture are solving? One would assume that the visual cortex is rather engaged in visual perceptual tasks, rather than planning and path finding.
- The objection for the type of task that is being investigated could arguably also apply to the use of ReNets as baseline models, since they have been built to solve visual perceptual tasks, while, based on the fact that Dijkstra's algorithm can be used to solve shortest path problems, one would assume that algorithms for path finding have more to do with implementing dynamic programming.
- The paper demonstrate that the novel proposed RNN module displays some interesting properties in how it seem to be tackling the path finding tasks differently from other RNNs and feeforward architectures. It is however not clear from the paper what computational or algorithmic features distinguish the different ways of tackling this class of problems. Presumably, the differences in performance have to do with the different inductive biases that different architectures have for solving this problem in the specific way that it is formulated (as visual classification tasks).

**Summary Of The Paper:**

The paper introduces a new gated convolutional recurrent architecture consisting of separate neural populations with long-range and short-range interactions. This structure is vaguely motivated by neuroanatomical cortical observations that revealed that neurons in the primary visual cortex of mammals are strongly recurrently coupled and this recurrence tends to be subdivided in long-range excitatory inputs and inhibitory inputs from local interneurons. In particular, these two pathways were implemented through convolutional operations with different kernel sizes, which are then mixed and combined with gating operations.
This novel recurrent architecture is then empirically compared against feed-forward vision architectures (ResNets) and recurrent vision architectures (ConvGRU and a recurrent version of ResNet obtained by tying weights across layers) on two previously proposed path integration tasks: Mazes (that asks to output a segment connecting two points in a 2d maze) and PathFinder (a classification tasks consisting on determining whether two disks on a image are connected or not by a curved segment among other distracting segments).
The proposed architecture performs better on these tasks than the competitors at most difficulty levels, although it is interestingly revealed that a version of ConvGRU is better at extrapolating on a difficult level after being trained on a different difficulty level. Related to this, ConvGRU is also shown to be more robust to varying the number of recurrent steps.

**Summary Of The Review:**

The paper starts with a interesting motivation, understanding the computational properties that the anatomy of cortex (high recurrence, separate local and long-range connection) provides to a RNN module. However, the paper doesn´t analyze the RNN module that is derived to incorporate these properties in a deep enough quantitative way. Moreover, the derived architecture is only benchmarked on a very restrictive set of tasks (path finding) that are arguably only a weak match with respect to the biological motivation (understanding cortex) and the conv-nets baselines.

---

> ### Author Response · Authors · 2022-11-19
> **Authors' response to comments from reviewer j482 (part 1)**
>
> Dear reviewer,
>
> We thank you for your time and thorough review of our submission. **We have uploaded a revised version of our submission during the revision phase with several new experiments and findings addressing reviewer comments.** For your concerns that overlapped with that of other reviewers, please find our detailed response in the common response thread above. We extensively improved our experiments with new baselines (Nayebi et al 2021, Kubilius et al 2018), we performed the ablation study that highlights your concern on bettering the mechanistic/algorithmic understanding of LocRNN and we also add a note with several important references that highlight both the relevance of our evaluated tasks to deep learning and to biological vision. Following is our response to your unique comments on our submission:
>
> - **On the mechanistic understanding of LocRNN:** Thank you very much for highlighting this key matter of bettering the mechanistic understanding of LocRNN and its individual components. We identify the computational and/or algorithmic features that contribute to LocRNN’s high performance on Mazes and PathFinder via an ablation study. We request you to please refer to part 2 of our common response for a detailed response to this concern.
> - **Static computational graph of feedforward networks:** We agree that feedforward networks with weight tying are computationally equivalent to shallow recurrent networks. However, **we argue that recurrent processing can alleviate the issue of feedforward networks possessing a fixed-length computational graph.
> Like in the brain, computation can be acted upon at arbitrary times via recurrent processing and this cannot be naturally mimicked by feedforward architectures.** Our recurrent networks can be operated for arbitrary number of timesteps whereas this is generally not the case for feedforward networks. We would like to also cite prior work on segmentation with recurrent networks that share our terminology of feedforward networks having a fixed computational graph [6]. However, we clarify our usage of the term fixed computational graph in our paper in Sec. 5.2.
> - **On the motivation for PathFinder/Mazes and their objection for the chosen tasks as visual tasks:** We respectfully disagree with your interpretation of PathFinder and Mazes as tasks that aren’t visual in nature and that they aren’t of interest for machine learning/deep learning. **Both PathFinder and Mazes evaluate the ability of deep learning models to integrate long-range spatial dependencies;** models take as input full RGB images as input and either perform classification (PathFinder) or segmentation (Mazes) which *require integrating spatially distant visual cues*.
> The biological relevance of both our evaluated tasks can be found in [5].
> PathFinder is a classic example of a curve tracing task performed by the visual cortex (see Fig. 5, 6, 7 in [5]) for contour integration performed by V1 neurons on stimuli very similar to what we use in our evaluation) and Mazes is a great adaptation of the serial grouping task in Fig 8 in [5]. Neither task involve exploring an environment for path finding; models receive per-pixel supervision for segmentation and not a scalar reward for finding a path like is usually the case in planning tasks. For a more detailed explanation of our stance on this issue, please refer to part 3 of our common response.

---

> > ### Author Response · Authors · 2022-11-19
> > **Authors' response to comments from Reviewer j482 (part 2)**
> >
> > (continuing response from part 1)
> > - **On the note of the relevance of Mazes and PathFinder to deep learning:** *Both Mazes and PathFinder were published in the context of deep learning research at NeurIPS 2018 and NeurIPS 2021 respectively*, and have been predominantly used so far in the context of deep learning research. These are certainly datasets that are of interest to the deep learning community and highlight certain key deficiencies of even today’s best-performing computer vision architectures (ViTs) as we mentioned in our common response to reviews. Both datasets are being used in the machine learning community to test deep learning models’ ability to learn long-range spatial dependencies [1, 2] and generalization [3, 4].
> >
> > **References:**
> > 1. Linsley, D., Kim, J., Veerabadran, V., Windolf, C., & Serre, T. (2018). Learning long-range spatial dependencies with horizontal gated recurrent units. Advances in neural information processing systems, 31.
> > 2. Tay, Y., Dehghani, M., Abnar, S., Shen, Y., Bahri, D., Pham, P., ... & Metzler, D. (2020). Long range arena: A benchmark for efficient transformers. arXiv preprint arXiv:2011.04006.
> > 3. Schwarzschild, A., Borgnia, E., Gupta, A., Huang, F., Vishkin, U., Goldblum, M., & Goldstein, T. (2021). Can you learn an algorithm? generalizing from easy to hard problems with recurrent networks. Advances in Neural Information Processing Systems, 34, 6695-6706.
> > 4. Bansal, A., Schwarzschild, A., Borgnia, E., Emam, Z., Huang, F., Goldblum, M., & Goldstein, T. (2022). End-to-end Algorithm Synthesis with Recurrent Networks: Logical Extrapolation Without Overthinking. arXiv preprint arXiv:2202.05826.
> > 5. Roelfsema, P. R. (2006). Cortical algorithms for perceptual grouping. Annual review of neuroscience, 29(1), 203-227.
> > 6. McIntosh, L., Maheswaranathan, N., Sussillo, D., & Shlens, J. (2018). Recurrent segmentation for variable computational budgets. In Proceedings of the IEEE Conference on Computer Vision and Pattern Recognition Workshops (pp. 1648-1657).

---

> > > ### Comment · Reviewer_j482 · 2022-11-23
> > > **Acknowledgment of authors' response**
> > >
> > > I would like to thank the authors for the effort in providing their rebuttals. The authors provided useful ablation studies that help in understanding the role of individual components in their proposed recurrent architectural module. This seems like a good firs step to indicate a quantitative analysis of the module in terms of exploding and vanishing gradients, a type of analysis which would arguably be helpful to provide formal guarantees of its stability and which for the moment is still missing.
> > > As for the choice of tasks, I agree with the authors that the PathFinder/Mazes tasks are of some interest within the community. My question regarding that point was rather about the evidence that these tasks are paradigmatic of the type of tasks that the visual cortex and its architecture are solving, since that hypothesis is one of the main motivations underlying the paper's proposal.
> > > The rebuttals provided by the authors also tried to address my objection against the statements that feedforward neural networks are "strictly restricted" compared to RNNs and "cannot dynamically change their computational graph". I noticed that these claims are still present in the introduction of the paper, despite the fact that the rebuttal seems to recognize that they should be qualified (at it's partially done later on in the paper). If anything, I think that this point following the rebuttal and the revisions is even more confusing. For one, now the rebuttals are confounding the stated incapability of feedforward networks to provide context-dependent representations (which they do actually can provide) with the possibility of unrolling an RNN for a fixed number of steps. My impression is that the rebuttals could have simply recalibrated those statements by recognizing that feedforward network are more general than RNN (in that their weights are not tied), can in fact provide context-dependent representations, and then separately considered the potential of being able to unroll a recurrent graph for an arbitrary number of steps. Instead, the rebuttal ended up muddling the conversation and making that point even more unclear, effectively nullifying and frustrating any attempt of my review to provide constructive criticism.
> > > All in all, I again want to thank the reviewers for their rebuttals. However, the rebuttals did not satisfactorily address or incorporate my comments. I will therefore maintain my score.

---

> > > > ### Author Response · Authors · 2022-11-29
> > > > **Replying to acknowledgement of authors' response**
> > > >
> > > > Thank you for responding to our rebuttal.
> > > > 1. We now more clearly understand the reviewer’s concern with our phrasing – we agree that feedforward networks are indeed a more general class of neural networks than RNNs and they do clearly provide context-dependent representations. The benefits of recurrent processing in task extrapolation through varying temporal iterations is a separate issue and the one our paper emphasizes and explores.  This task extrapolation through increased processing is similar to humans exhibiting speed/accuracy tradeoffs and taking longer on harder tasks (e.g. multiplying 3 digit vs 2 digit numbers)).
> > > >
> > > >     We apologize for conflating these issues in our Introduction and Discussion and will separate these out as the reviewer has suggested in their response to our rebuttal.
> > > >
> > > > 2. We thank you for recognizing the many ablation analyses that we performed to better understand the components of LocRNN that are critical for improved performance.
> > > >
> > > > 3. On vanishing and exploding gradients: We thank you for this constructive comment and shall take this feedback to explore the gradient behavior of LocRNN in future scope. However, while we agree that this analysis is relevant to the proposed work, it is not central to our paper’s message that specialized recurrent networks can flexibly adapt computation to exhibit task extrapolation on visual contour integration tasks. As suggested by the reviews in general, we feel that our experiments do strongly convey this interesting finding and believe this will stimulate further research within the ICLR community.
> > > >
> > > > 4. “My question regarding that point was rather about the evidence that these tasks are paradigmatic of the type of tasks that the visual cortex and its architecture are solving, since that hypothesis is one of the main motivations underlying the paper's proposal.”
> > > > Here we wish to re-emphasize that these tasks are indeed very much paradigmatic of the type of tasks that the visual cortex and its architecture are solving as evidenced by the neuroscience [e.g. 5, 7] and behavioral [e.g. 2, 3, 7] research and prior work modeling the visual cortical response to similar stimuli [e.g. 6]. Hence, we also maintain the choice of ResNets and R-ResNets as baselines are justified as these tasks are indeed visual in nature (image classification and segmentation).
> > > >
> > > > References:
> > > > 1. Tay, Y., Dehghani, M., Abnar, S., Shen, Y., Bahri, D., Pham, P., ... & Metzler, D. (2020). Long range arena: A benchmark for efficient transformers. arXiv preprint arXiv:2011.04006.
> > > > 2. Jolicoeur, P., & Ingleton, M. (1991). Size invariance in curve tracing. Memory & Cognition, 19(1), 21-36.
> > > > 3. Roelfsema, P. R., & Houtkamp, R. (2011). Incremental grouping of image elements in vision. Attention, Perception, & Psychophysics, 73(8), 2542-2572.
> > > > 4. Ullman, S. (1987). Visual routines. In Readings in computer vision (pp. 298-328). Morgan Kaufmann.
> > > > 5. Li, W., Piëch, V., & Gilbert, C. D. (2008). Learning to link visual contours. Neuron, 57(3), 442-451.
> > > > 6. Li, Z. (1998). A neural model of contour integration in the primary visual cortex. Neural computation, 10(4), 903-940.
> > > > 7. Roelfsema, P. R. (2006). Cortical algorithms for perceptual grouping. Annual review of neuroscience, 29(1), 203-227.

---

### Official Review · Reviewer_i8Ub · 2022-11-04

**Confidence:** 3
**Correctness:** 3
**Technical Novelty And Significance:** 3
**Empirical Novelty And Significance:** 3
**Recommendation:** 6

**Clarity, Quality, Novelty And Reproducibility:**

**Clarity**
- Results section is claims-driven, which is good, but could overall use more signposting. It does help to have certain sentences bolded.
- Figures, particularly 2 and 3, are whitespace-heavy and lack clear highlights of important trends
- Some analysis isn't fully fleshed out. For example, the connection between Fig 3B (connection between validation accuracy as a function of recurrent iterations) and stability seems fair, but I would benefit from clearer explanation of how to interpret the graphical results and what specifically they mean for stability

**Quality**
- Main issue is need for more experimentation
- Quality of experiment execution seems high

**Originality**
- This paper dos not compare to Nayebi et al. 2021 (ConvRNN), which also uses long connections (though not lateral). It is the most similar work I know of and it would help for the authors to address it

**Strength And Weaknesses:**

**Strengths**
- Architecture is intuitive and appears to be a simple but effective model and discretization of the human neural "long lateral connections"
- Paper is well-scoped - it raises certain questions and answers them effectively with experiments that tie into each other
- Experiments are well-thought out and support claims
- Results are significant, particularly the extrapolation result that shows good performance on
- Generality of approach, compared to more specialized architectures, is exciting

**Weaknesses**
- Experimental suites are somewhat limited, could benefit from more tasks
- Not enough results are presented in the main paper - we are told that LocRNN has better task performance than others, but there are no tables to highlight the statistics and Figure 2 doesn't clearly demonstrate this relative to GRU (or at least, I didn't notice)
- The tradeoff section is unconvincing - it's a brief paragraph relying on a visualization, and even if it had more concrete evidence, the connection between the statistics presented and hidden state stability is unclear. Ultimately, we have three data points, not an empirically established or theoretically proven trend (as far as the paper goes), though connections could be established.

**Summary Of The Paper:**

This paper presents a recurrent convolutional architecture (LocRNN) to improve extrapolation in visual tasks. The novelty of the architecture is presented as being long-range lateral connections between neurons from the same layer. The architecture is tested on two tasks: Mazes (route segmentation) and PathFinder (curve tracing). Both of these have multiple difficulty levels to test on.

LocRNN architecture is based on an ODE-based model of lateral connections between cortical columns in the primate brain. The architecture uses a long range neural population and a short range neural population, each of which has gated updates and lateral connections defined by convolutional kernels within and between the neural populations.

This is added to residual network architecture, and compared with GRU and ResNets themselves. LocRNN and GRU outperform ResNets (feedforward only) on both tasks. To show extrapolation, experiments are conducted by training on one difficulty level and testing on the rest - ConvGRU and LocRNN both do well in extrapolation, and ConvGRU (specialized to sequential processing) does the best.

Finally, the paper claims that extrapolation ability (as shown by the existing recurrent architectures) trades off with task performance because performant algorithms lack stability in the hidden space.

**Summary Of The Review:**

I am recommending a weak acceptance because I do not see any major flaws in the paper and the general agent performance results support the main claims they are attached to. I am not recommending strong acceptance because important and very similar convolutional recurrent networks prior work has not been compared to, and the scope of the experiments and method is somewhat limited. Furthermore, the tradeoff is addressed briefly despite being a main contribution and a compelling claim, and the paper would benefit from doing more for that claim.

---

> ### Author Response · Authors · 2022-11-19
> **Authors' response to comments from Reviewer i8Ub**
>
> Dear reviewer,
>
> We thank you for your time and careful evaluation of our paper. **We have uploaded a revised version of our submission during the revision phase with several new experiments and findings addressing reviewer comments.** We request you please carefully consider our common response above. It addresses an important and valid issue you raised in our paper, of evaluating the very relevant ConvRNN model from Nayebi et al 2021. We have evaluated ConvRNN as well as the CORNet-S model (Kubilius et al 2018) and found very intriguing results on raw performance and extrapolation. Additionally, we also include an ablation study to our paper that better provides a mechanistic understanding of LocRNN and we provide a key explanation of the relevance of our dataset to biological vision and the level of difficulty involved in performing the tasks we experiment with. Following is our response to your unique comments on our paper:
>
> - **“Experimental suites are somewhat limited, could benefit from more tasks”** – As per your feedback and that of vnpB, we will be performing experiments evaluating LocRNN’s performance on new tasks in future scope. However, we find these new experiments to be out of the scope of our current submission where our goal is to estimate the ability of deep learning models to integrate long-range spatial dependencies and to perform task extrapolation.
> Adding another task significantly increases our computation cost; we train between 6-9 baseline models (w/ 3 random seeds) on 4 kinds of experiments (evaluation on Mazes, PathFinder, task extrapolation on Mazes, PathFinder). Adding another task to this pipeline would mean 2 more additional kinds of experiments on all 6-9 baselines w/ 3 random seeds which can be highly expensive to run.
> As noted by Reviewer 3CSc, we believe the following statement they make is accurate “the choice of datasets as well as models to compare with is good.”. For the context of our questions addressed in this paper, we request you please reconsider the additional requirement of more tasks.
> - **On the performance of LocRNN** – We would like to highlight that we do not claim better performance of LocRNN on Mazes.
> Quoting from our paper’s Sec 5.1 : “However we show that both ResNets and R-ResNets are considerably different from all specialized recurrent architectures (LocRNN, hConvGRU, ConvGRU) which are all more accurate and sample efficient compared to the ResNets. We would like to emphasize that this difference is significant and argues for improved effectiveness of specialized recurrent architectures..” We have further emphasized this in our paper and attributed ConvGRU for its superior performance on Mazes. **However, we would like to suggest taking this result in combination with results on PathFinder where ConvGRU is at chance on all difficulty levels!**. On PathFinder, LocRNN is the best performing architecture considering performance of each model at each difficulty level, and we emphasize this in the paper.
> - **More signposting** – Thank you for your advice for us to use more signposting. Certainly, we have incorporated this in our revision with further highlights of our main claims and supporting evidence.
> - **On the stability vs expressivity analysis** – Thank you for your interest in this very important contribution we make in this work. We note your request for a better explanation of stability as we discuss it in the paper; we refer to stability from the perspective of stability of dynamical systems. LocRNN being a dynamical system characterized by difference equations, we address LocRNN as stable if it is capable of mapping input images to a fixed point/attractor where the network dynamics are in equilibrium. We have further clarified this point in our paper in Sec. 5.4.

---

### Author Response · Authors · 2022-11-16
**Author response addressing common points from all reviewers and AC (part 1, experimenting more baselines)**

Dear Reviewers and AC,

We thank you for your time and careful consideration of our paper. This response addresses common points. We shall provide briefer comments separately to each reviewer to address their unique comments. We have updated our manuscript during this rebuttal phase and our *references to figures in this author response correspond to the latest version of our manuscript*.

**We provide our response to common points in 3 parts due to character limit. This is part 1; part 2 and part 3 of this response can be found below in the same thread.**

First, we are grateful for the kind and positive feedback provided by our reviewers. We are encouraged that reviewers find the problem of task extrapolation using biologically plausible recurrent models to be of high interest and impact. We are glad that multiple reviewers:
- find our proposed LocRNN architecture to be intuitive and simple, yet effective with generality and high performance (i8Ub, vnpB, 3CSc, j482).
- find the experiments to be thorough and supportive of our claims (i8Ub, vnpB, 3CSc).
- are keen on the proposed tradeoff between task performance and stability and would like to see further experimentation in this section (vnpB).

In line with the reviewers’ constructive feedback on our work, we have performed several key extensions to our analyses that we have included in our current revision. We hope these additional observations **on LocRNN’s supremacy over new relevant recurrent convolutional baselines, ablation showing the importance of interneurons, ReLU and other model components, improved expressivity of GRU with ReLU, etc.** which are highly intriguing in our view provide convincing evidence for our reviewers and ACs to unanimously find the paper as relevant and important for the ICLR audience. We organize our rebuttal into the following subsections:

## Adding more baseline convolutional recurrent networks (Also see updated Fig 1).
tl;dr: We added 2 more new baseline convolutional recurrent networks to our benchmarking on PathFinder and Mazes: ConvRNN (requested by i8Ub) and the relevant CORNet-S. **LocRNN outperforms both new models in terms of accuracy and sample efficiency**

We see the reviewers’ intent that our previous comparisons did not include certain key similar models of lateral connections from prior work. *We agree with you and address this comment with performance analyses of 2 more important baselines explained below.*
Particularly, as reviewer i8Ub pointed out, we agree that ConvRNN is similar to our proposed LocRNN model; however, we would like to highlight that there are certain key differences in the way that gating and lateral connections are implemented in these two networks. These differences between LocRNN and ConvRNN are reflected in their performance differences.

**Neither ConvRNN nor CORNet-S learns all PathFinder difficulty levels.**

While matching the base architecture, receptive field size, and the number of timesteps of recurrence between ConvRNN and other RNNs used in our evaluation, we observe that *ConvRNN cannot effectively learn any of the PathFinder challenges in the given number of epochs.* Whereas, LocRNN learns the task in less than one epoch.
ConvRNN is only capable of converging on 1 out of 3 initializations on PathFinder-9 and does not converge on PathFinder-14/18. (See updated Fig 1)
On the other hand, *CORNet-S is slightly better than ConvRNN in that it converges effectively on PathFinder-9.* However on PathFinder-14 and PathFinder-18, CORNet-S converges only on 1 out of 3 random initializations respectively.
*This behavior is inferior to LocRNN which consistently converges on 3 random initializations of all 3 difficulty levels.* To clarify, we choose 3 seeds randomly and use it for all model’s random initializations. *THEY ARE NOT chosen to optimize LocRNN’s performance.*

**We have updated Figure 1 with the latest evaluations of ConvRNN and CORNet-S and request the reviewers to please check this. The fourth and fifth panels in the bottom part of the figure are simulations of the new models.**

**Note on LocRNN used for Mazes:** During the revision phase we realized that we had used a version of LocRNN for Mazes that had significantly fewer (0.5x) parameters than other recurrent networks compared. We corrected this by training a version of LocRNN that is parameter-matched to other recurrent network baselines. Our revision shows the performance of this parameter-matched LocRNN that exhibits better raw performance and generalization than the LocRNN reported in our submission prior to revision.

---

> ### Author Response · Authors · 2022-11-16
> **Author response addressing common points from all reviewers and AC (part 2, understanding LocRNN / ablation study)**
>
> ## Better understanding LocRNN’s components – ablation study (Also see Fig 3B).
> tl;dr: Ablation study reveals the relative importance of the various LocRNN components. Importance quantitatively evaluated as drop in performance on PathFinder-14. Overall, the whole LocRNN circuit performs better than its lesioned counterparts.
>
> We thank the reviewers for raising the request to better understand LocRNN’s components / to perform an ablation of LocRNN. The following are LocRNN’s individual components that we systematically lesion in this experiment: 1) Interneuron property of S population (2 populations of neurons vs 1 population), 2) Non-saturating ReLU nonlinearity, 3) Gating and 4) LayerNorm.
> We present results from lesioning each of these individual components of LocRNN and evaluating the resulting models on PathFinder-14. We choose PathFinder-14 to run this ablation study as it is the dataset that starts showing individual performance differences among architectures.
> For clarity, we remove individual components from LocRNN and train the resulting network whose performance is then evaluated on the validation set.
>
> **Interneurons are crucial for good performance:** Our network uses L and S neurons that differ in lateral connectivity extent, but also importantly in whether they project to the output. We call the S neurons that don’t project to the output interneurons (http://www.scholarpedia.org/article/Interneurons).
> We found strong evidence suggesting that interneurons are crucial to LocRNN’s high performance. First, we retained L neurons and removed S neurons in LocRNN. This ablation converges to a stable solution in only 1 out of 3 random seeds (also with a *~38% drop in sample efficiency*). Next, to match the number of parameters, we increased the width of L neurons to 64 channels. While there was a slight improvement (2 out of 3 seeds converged), the sample efficiency of the models that converged was still very low compared to the full model (25% drop in sample efficiency).
> In an interesting third ablation experiment, we **retained both L and S neurons and sent both neuron populations upward** for classification. This explicitly tests whether the interneuron property helps. Again, only one of the 3 seeds converged to an accurate solution and with very poor sample efficiency, highlighting that the property of S neurons being **interneurons is of utmost importance for LocRNN’s high performance!**
>
> **ReLU is important for good performance:  Having a long range of linear (non-saturating) operating mode in the activation function is critical.  Replacing ReLU
> with tanh** completely breaks LocRNN’s performance. However, replacing ReLU with a clipped ReLU (clipped at 6) did not cause any drop in performance. Also, in agreement with this, when we replace tanh in ConvGRU with ReLU, we achieve a significant gain in ConvGRU’s PathFinder-14 performance (+10.4%).
>
> **Gating and LayerNorm contribute to improving performance and sample efficiency:** We find that LayerNorm affects performance the least among all components. Even in the absence of LN, LocRNN achieves >80% performance on PathFinder. However, LN seems to improve sample efficiency significantly, similar to gating the absence of which hurts both performance and sample efficiency.

---

> > ### Author Response · Authors · 2022-11-16
> > **Author response addressing common points from all reviewers and AC (part 3, on task complexity and relevance to biological vision)**
> >
> > ## On the note of dataset simplicity and relevance to biology:
> > We note that some of our reviewers (j482, vnpB) find our tasks to either be too simple or irrelevant to visual cortical processing. We respectfully disagree with this and explain our stance below.
> >
> > **Vision Transformer fails to learn PathFinder:** We would like to note that Vision Transformers are making huge strides in the field of computer vision today and are a strong baseline for several visual tasks. Here we evaluated a 6-layer deep vision transformer with ~10M parameters (2 orders larger than LocRNN) on PathFinder-14. Across 3 random initializations, **none of the ViT models converged to an accurate solution** and all models were at chance. This is one strong case highlighting that the **evaluated tasks are indeed challenging benchmarks and highlight key deficits of current state-of-the-art deep learning models**.
> > Our result echoes the findings of [1] where the authors also find that Transformers are unable to solve PathFinder. [1] rasterized their input images but here, we respect the structure of 2D images using ViTs; in both cases, ViTs fail to learn PathFinder.
> >
> > We would like to emphasize how **the failure of multiple models** (hConvGRU, ConvGRU, ConvRNN, CORNet-S, ResNet-10, ViT) of varying depths and # of parameters on multiple difficulty levels of PathFinder and Mazes **suggests that these datasets are challenging** despite the apparent transparency of the tasks. PathFinder, one of the tasks we use is also part of a large benchmark (Long Range Area, [1]) used to evaluate the ability of various transformers in integrating long-range dependencies.
> >
> > **Important point of clarification:**
> > The comments of reviewer j482, made us realize, that we should emphasize that these tasks are not higher-order planning tasks; all information is available in the input and just needs to be integrated across the receptive fields - there are no state-exploring actions needed. Per stimuli used in prior cognitive psychology and behavioral research [2, 3, 4], these are visual perceptual (contour integration/curve tracing/segmentation) tasks that are considered to be computed in visual cortical areas and likely rely on lateral connections (which our model features) to integrate the contours [5, 6]. [7] shows how lateral interactions modulate neuronal responses in PathFinder-style stimuli. We apologize that the “PathFinder” and “Mazes” names (invented by [8] and [9]) obscure this critical issue.
> >
> > **References**
> > 1. Tay, Y., Dehghani, M., Abnar, S., Shen, Y., Bahri, D., Pham, P., ... & Metzler, D. (2020). Long range arena: A benchmark for efficient transformers. arXiv preprint arXiv:2011.04006.
> > 2. Jolicoeur, P., & Ingleton, M. (1991). Size invariance in curve tracing. Memory & Cognition, 19(1), 21-36.
> > 3. Roelfsema, P. R., & Houtkamp, R. (2011). Incremental grouping of image elements in vision. Attention, Perception, & Psychophysics, 73(8), 2542-2572.
> > 4. Ullman, S. (1987). Visual routines. In Readings in computer vision (pp. 298-328). Morgan Kaufmann.
> > 5. Li, W., Piëch, V., & Gilbert, C. D. (2008). Learning to link visual contours. Neuron, 57(3), 442-451.
> > 6. Li, Z. (1998). A neural model of contour integration in the primary visual cortex. Neural computation, 10(4), 903-940.
> > 7. Roelfsema, P. R. (2006). Cortical algorithms for perceptual grouping. Annual review of neuroscience, 29(1), 203-227.
> > 8. Linsley, D., Kim, J., Veerabadran, V., Windolf, C., & Serre, T. (2018). Learning long-range spatial dependencies with horizontal gated recurrent units. Advances in neural information processing systems, 31.
> > 9. Schwarzschild, A., Borgnia, E., Gupta, A., Huang, F., Vishkin, U., Goldblum, M., & Goldstein, T. (2021). Can you learn an algorithm? generalizing from easy to hard problems with recurrent networks. Advances in Neural Information Processing Systems, 34, 6695-6706.

---

### Decision · Program_Chairs · 2023-01-20

**Decision:**

Reject

**Justification For Why Not Higher Score:**

1) Overly general claims that are not well-supported by results
2) Weak contributions, as least in terms of articulation

**Justification For Why Not Lower Score:**

n/a

**Metareview: Summary, Strengths And Weaknesses:**

This paper introduces a novel recurrent neural network model (LocRNN) that has biological motivation and adherence, such as long-range and short-range interactions. The model is tested on task extrapolation, in which the level of computation required to solve a problem can be dynamically altered depending on the task's difficulty. Specifically, the model is tested on the Mazes and Pathfinder benchmark tasks (both related to the contour integration and curve tracing capabilities found in visual brain areas) with several levels of difficulty, and somewhat outperforms other baselines. The experiments support the claims that the proposed model is able to perform task extrapolation (training on one difficulty level and testing on others).

-- STRENGTHS --

1) Technical novelty of proposed method, especially with fair amount of proper biological motivation

2) Generally robust evaluation and clear results, especially after revision by authors in response to initial reviews.


-- WEAKNESSES --

(A good number of discussion points about weaknesses on OpenReview were more about points needing clarification, and they were pretty technical or specific. Also, the authors managed to address a good number of reviewer concerns, and the reviewers upgraded their scores. Hence, here I focus on key big-picture weaknesses, which I thought were somehow not adequately covered by the reviews.)

Overall, the claimed contributions, while consistent with the results, are either weak contributions or are only weakly supported. One example is the title, which makes a very general statement about recurrence enabling task extrapolation. While the proposed model may indeed show this, the phrasing of the title suggests that all or any form of recurrence could enable task extrapolation, which certainly has not been shown. More specific examples are as follows.

1) Claimed contribution #1, that the paper shows the advantage of recurrent over feedforward processing on task extrapolation, is only weakly supported. The paper ultimately only tests a small number of models on two datasets. No doubt any single paper can only perform a limited number of experiments, but then surely the claim should then be scoped down accordingly. As a reviewer pointed out, the claim would be much more convincing if theoretical proofs or detailed mechanistic explanations were provided.

2) As stated in the paper, claimed contribution #2 is simply that of the LocRNN model, which purportedly introduces a new type of connection, which is hypothesized to enable learning of visual routines. This is a very weakly worded contribution. While one could counter-argue that this is simply a matter of writing, my view is that this is symptomatic of the paper overall, which is not clear on what exactly is the "story" it is trying to tell.

3) Claimed contribution #3 is that the paper *posits* "the potential tradeoff between task performance ... vs stability...". This is a very weak thing to claim as a contribution. To just posit (put forward) an idea that is potentially true, based simply on "an instance of this tradeoff we observe" is not really a proper contribution in my view, but rather an observation more suitable as an interesting point for the discussion section.


-- OVERALL --

Overall, this is not a bad paper by any means. The model is fairly novel and interesting, the results are positive, and I personally like the proper treatment given to the biologically-inspired approach. However, the reviewers are all lukewarm at best about this paper, with hardly any real excitement or enthusiasm even when describing the strengths. These are reflected in both their qualitative and quantitative feedback.

In making my recommendation (to reject), I took a big-picture view of the paper: what is the real significance and contribution of this work, as written by the authors? The authors chose to explicitly claim scientific-type contributions, and wrote the abstract, intro and discusson as such (in a more typically scientific style rather than AI/ML style), even though the body of the paper is a pretty typical dataset-model-evaluation-analysis one. I found the paper to be very weak in terms of articulating accurately and precisely how the community would benefit from this work, with vague or hedged phrases such as "our work encourages further study", "we show promising evidence", etc.

I find the work to be a tad too preliminary for acceptance in its current form. However, the core of the paper holds potential for some really interesting contributions (e.g. "we show that a recurrent network is able to flexibly adapt its computational budget during inference"), and I would encourage the authors to continue this line of work and keep improving it.


(Note: the authors are not wrong to point out that reviewer j482's score of 3 may not be commensurate with the content of their review, as well as their subsequent response to the rebuttal. I have factored this into my recommendation).



**Summary Of Ac-Reviewer Meeting:**

n/a